# Implementing the Reactor Geometry in the Modeling of Chemical Bath Deposition of ZnO Nanowires

**DOI:** 10.3390/nano12071069

**Published:** 2022-03-24

**Authors:** Clément Lausecker, Bassem Salem, Xavier Baillin, Vincent Consonni

**Affiliations:** 1Université Grenoble Alpes, CNRS, Grenoble INP, LMGP, 38000 Grenoble, France; clement.lausecker@grenoble-inp.fr; 2Université Grenoble Alpes, CNRS, CEA/LETI-Minatec, Grenoble INP, LTM, 38054 Grenoble, France; 3Université Grenoble Alpes, CEA, LETI, 38054 Grenoble, France; xavier.baillin@cea.fr

**Keywords:** ZnO nanowires, chemical bath deposition, predictive modeling

## Abstract

The formation of nanowires by chemical bath deposition is of great interest for a wide variety of optoelectronic, piezoelectric, and sensing devices, from which the theoretical description of their elongation process has emerged as a critical issue. Despite its strong influence on the nanowire growth kinetics, reactor size has typically not been taken into account in the theoretical modeling developed so far. We report a new theoretical description of the axial growth rate of nanowires in dynamic conditions based on the solution of Fick’s diffusion equations, implementing a sealed reactor of finite height as a varying parameter. The theoretical model is applied in various chemical bath deposition conditions in the case of the growth of ZnO nanowires, from which the influence of the reactor height is investigated and compared to experimental data. In particular, it is found that the use of reactor heights smaller than 2 cm significantly decreases the ZnO nanowires’ axial growth rate in typical experimental conditions due to the faster depletion of reactants. The present approach is further used predictively, showing its high potential for the design of batch reactors for a wide variety of chemical precursors and semiconductor materials in applied research and industrial production.

## 1. Introduction

Chemical bath deposition (CBD) has received increasing interest in the last two decades as a low-cost and low-temperature process to form nanostructured materials such as semiconducting chalcogenides and oxides [1,2,3] used in a broad range of applications in the fields of optoelectronics and photovoltaics [4,5,6], piezoelectricity [7,8,9], and sensing [10]. It is further easily implemented on a variety of flexible and rigid substrates and can be scaled up for industrial purposes [11]. CBD of ZnO nanowires (NWs) is a particularly popular example of the use of this technique, which has given rise to an extensive literature [11,12,13,14,15,16,17,18]. A simply closed or sealed reactor containing an aqueous solution of the chemical precursors composed of a zinc salt and an amine is typically heated up to a temperature in the range of 60–90 °C, resulting in the spontaneous formation of ZnO in the form of NWs [19,20,21,22]. A dedicated nucleation surface is further introduced into the chemical bath to achieve the heterogeneous formation of ZnO NWs and is usually in the form of a ZnO polycrystalline seed layer [13,23,24], a ZnO monocrystal [25,26], or a metallic seed layer [27,28,29]. The use of a simply closed or sealed reactor with no further addition of chemical precursors during the CBD process has gained broad interest due to the relative simplicity of this system [15,16,20,22,28,29]. In this configuration, the chemical bath is progressively depleted in reactants as the CBD process proceeds, especially in Zn(II) ions as the limiting species owing to their consumption to form ZnO NWs. As a result, ZnO NWs grow under dynamic conditions, where their axial growth rate typically decreases over time. The growth kinetics of ZnO NWs is highly dependent upon (i) the chemical bath characteristics [21,22,27,28], namely its temperature and the precursor concentrations, and (ii) the reactor size, which defines the total amount of precursors available. The elongation of ZnO NWs and its temporal dependence has, thus, emerged as a critical issue for applied research and industrial production where high throughput and precise control over the NW length are usually required.

In this context, the theoretical modeling of the elongation process of ZnO NWs by CBD deserves particular attention [30,31,32,33,34]. However, the expression of their axial growth rate has mainly been limited to static conditions, in which the concentration profile of the reactants in the chemical bath is considered constant with time, and thus, its temporal dependance is neglected [30,31,32]. While this approach can be used in specific conditions under which the depletion in reactants is negligible—e.g., in a continuous-flow reactor, it is basically not suitable to describe the most common configuration of a closed reactor subjected to dynamic processes. From this approach, Boercker et al. demonstrated that the length of ZnO NWs is inversely proportional to its surface density in a growth regime limited by the diffusive transport of chemical reactants [30]. Cheng et al. developed a continuous-flow reactor maintaining homeostatic chemical precursor concentrations and performed a comprehensive analysis of the surface reaction-/diffusive transport-limited growth regimes [31]. They further established the expression of the axial growth rate, *R_c_^stat^*, in homeostatic conditions, from which the elongation of ZnO NWs, *L^stat^*, is deduced as follows:(1)$$L^{stat}=R_{c}^{stat}t=\frac{k_{1}C_{0}D}{\rho D+\rho \delta k_{1}S}t$$
where *C*_0_ is the concentration of Zn(II) ions (m^−3^), *k*_1_ is the first-order reaction rate constant describing the crystallization process of ZnO (m·s^−1^), *D* is the diffusion coefficient of Zn(II) ions in aqueous solutions at a given temperature (m^2^·s^−1^), *S* is the *c*-plane top surface area ratio defined by the area of the top surfaces of all *c*-plane ZnO NWs divided by a given substrate surface area, *ρ* is the atomic density of wurtzite ZnO that is equal to 4.20 × 10^28^ m^−3^, *δ* is the stagnant layer thickness subjected to diffusive transport (m), and *t* is the growth time (s). Cossuet et al. subsequently applied this expression to the growth of ZnO nanorods of different polarities selectively grown on ZnO single crystals, where the active area was sufficiently small compared to the volume of solution to consider static conditions [32]. They showed that Zn-polar ZnO nanorods had a higher growth rate compared to O-polar ZnO nanorods owing to their larger surface reaction rate constant *k*_1_. Later on, a novel theoretical model introducing dynamic conditions, and hence compatible with the use of a closed reactor subjected to no macroscopic convection, was established by solving the Fick’s diffusion equations [33]. The expressions of the time-dependent axial growth rate of ZnO NWs, *R_c_^dyn^*, and of their elongation, *L^dyn^*, are formulated as follows:(2)$$R_{c}^{dyn}\left(t\right)=\frac{(C_{0}-C_{eq})k_{1}}{\rho }\left[\exp\left(\frac{{\left(k_{1}S\right)}^{2}}{D}t\right)\mathrm{erfc}\left(\frac{k_{1}S}{\sqrt{D}}\sqrt{t}\right)\right]$$
(3)$$L^{dyn}\left(t\right)=\frac{(C_{0}-C_{eq})D}{\rho k_{1}S^{2}}\left[\exp\left(\frac{{\left(k_{1}S\right)}^{2}}{D}t\right)\mathrm{erfc}\left(\frac{k_{1}S}{\sqrt{D}}\sqrt{t}\right)+2k_{1}S\sqrt{\frac{t}{D\pi }}-1\right]$$
where *C_eq_* is the equilibrium concentration of Zn(II) ions (m^−3^). Using a similar approach, Černohorský et al. employed finite element method calculations to solve the Fick’s diffusion equation in two dimensions and assessed the impact of the transverse diffusion of chemical reactants on the growth of small arrays of ZnO NWs [34]. However, these different studies have systematically considered the reactor as an infinite reservoir of chemical reactants, and the influence of the reactor size on the elongation process of ZnO NWs has basically not been investigated yet despite its critical importance.

In this article, we develop a new theoretical model of the time-dependent axial growth rate of ZnO NWs by CBD under dynamic conditions, where the reactor size is implemented through the consideration of a finite reactor height. Its influence on the axial growth rate is investigated for a broad range of CBD conditions and validated by experimental data. The evolution of the elongation of ZnO NWs from this approach is further compared to the theoretical model developed in our previous work considering a semi-infinite reactor [33], from which typical conditions yielding to maximized axial growth rates were deduced. The present theoretical model shows great potential to predictively describe the CBD process of NWs from a limited volume of solution and can be adapted to a wide variety of chemical reactants and semiconductor materials for the design of batch reactors in applied research and industrial production.

## 2. Materials and Methods

### 2.1. Deposition Techniques

The samples were prepared from 2 × 2 cm^2^ Si (100) substrates (Si-Mat) cleaned with acetone and isopropyl alcohol in an ultrasonic bath. To favor ZnO nucleation from the substrate during the CBD process, a polycrystalline ZnO seed layer was deposited by dip coating from a solution containing 375 mM of zinc acetate dihydrate (Zn (CH_3_COO)_2_·2H_2_O, Sigma-Aldrich, St. Louis, MO, USA) and 375 mM of monoethanolamine (MEA, Sigma-Aldrich, St. Louis, MO, USA) in pure ethanol (Fisher Scientific, Merelbeke, Belgium, 99.8% purity). The substrates were dipped into the solution under a controlled atmosphere (air with <15% hygrometry) and subsequently annealed for 10 min at 300 °C to evaporate residual organic compounds for 1 h at 500 °C to crystallize the polycrystalline ZnO seed layer.

ZnO NWs were synthesized by CBD using a reactor sealed with a piece of glass coated with Parafilm and containing an aqueous solution (Milli-Q water, 18.2 Ω·cm) of zinc nitrate hexahydrate (Zn(NO_3_)_2·_6H_2_O, Sigma-Aldrich, St. Louis, MO, USA) and hexamethylenetetramine (HMTA, C_6_H_12_N_4_, Sigma-Aldrich, St. Louis, MO, USA) in equimolar concentrations of 30 mM. To tune the reactor size by obtaining different heights of chemical bath ranging from 0.5 to 4.0 cm, the total volume of solution was varied from 4.6 to 37.4 mL. Each sample was maintained face down at the top surface of the chemical bath using Kapton, as depicted in Appendix A. No stirring was performed to favor diffusion processes during the CBD. The sealed reactors were placed in a regular oven heated at 90 °C for 41 h.

### 2.2. Characterization Techniques

The morphological properties of ZnO NWs were investigated with an FEI Quanta 250 field-emission scanning electron microscopy (FESEM) instrument (FEI, Hillsboro, OR, USA). For each sample, the mean length was deduced from cross-sectional-view FESEM imaging over a population of more than 60 NWs, with an associated error margin corresponding to the standard deviation obtained. By using ImageJ software (version 1.44p), the *c*-plane top surface area ratios, *S*, of ZnO NWs were inferred from top-view FESEM imaging, in which a filter was applied to only reveal their *c*-plane top facets.

## 3. Results

### 3.1. Description of the Theoretical Model under Dynamic Conditions

To implement the reactor geometry in the modeling of the elongation process of ZnO NWs, we consider a reactor of finite height, *h*, defined such that 0 < *z* < *h*, where an infinite substrate is placed at *z* = 0, as depicted in Figure 1. To establish the expression for the length of ZnO NWs under dynamic conditions—i.e., in which the depletion of chemical reactants occurs, we apply Fick’s second diffusion equation in one dimension on the concentration of Zn(II) ions [35]:(4)$$\frac{\partial C}{\partial t}=D\frac{\partial^{2}C}{\partial z^{2}}$$
with the three following boundary conditions:(5)$$C\left(z,t=0\right)=C_{0}$$
(6)$$\frac{dC\left(z=h,t\right)}{dz}=0$$
(7)$$\frac{dC\left(z=0,t\right)}{dz}=\frac{k_{1}S}{D}\left(C\left(z=0,t\right)-C_{eq}\right)$$
where *C* is the concentration of Zn(II) ions (m^−3^) at a height *z* above the substrate and at a growth time *t*, *C_eq_* is the equilibrium concentration of Zn (II) ions (m^−3^), *k*_1_ is the first-order reaction rate constant describing the crystallization process of ZnO (m·s^−1^), *S* is the *c*-plane top surface area ratio, and *D* is the diffusion coefficient of Zn (II) ions in aqueous solution at a temperature *T* (m^2^·s^−1^).

Equation (5) reports that the concentration of Zn(II) ions at a height *z* and at *t* = 0 is assumed constant at any point in the chemical bath and is defined as *C*_0_. Equation (6) accounts for the finite height of the reactor. It is derived from Fick’s first diffusion equation by assuming that the flux of Zn(II) ions at *z* = *h* equals 0. Equation (7) relates to the consumption of Zn(II) ions on the growth front located at the *c*-plane top facet of ZnO NWs. It is obtained by applying Fick’s first diffusion equation at *z* = 0, while considering Zn(II) ions as the limiting reactants. It is also worth noticing that the development of the growth front with time along the *z* axis is neglected and that *S* is considered as a constant parameter in a first approximation.

To solve this second-order partial differential equation, we apply the Laplace transform to Equation (4). By further using Equation (5), we obtain the following:(8)$$\frac{\partial^{2}c}{\partial z^{2}}-\frac{p}{D}c=\frac{-C_{0}}{D}$$
where *c* (*z*,*p*) is the Laplace transform of *C* (*z*,*t*). The temporal derivative is, thus, removed, and we obtain an ordinary differential equation. By using Equations (6) and (7), Equation (8) can be solved, and we obtain the following:(9)$$c\left(z,p\right)=\frac{C_{0}}{p}-\frac{\left(C_{0}-C_{eq}\right)}{p}\left[\frac{cosh\;\left(\sqrt{\frac{p}{D}}\left(h-z\right)\right)}{cosh\left(\sqrt{\frac{p}{D}}h\right)+\frac{\sqrt{pD}}{k_{1}S}sinh\left(\sqrt{\frac{p}{D}}h\right)}\right]$$

It is worth noticing that at *z* = 0, Equation (9) simplifies into
(10)$$c\left(0,p\right)=\frac{C_{0}}{p}-\frac{\left(C_{0}-C_{eq}\right)}{p}\frac{1}{1+\frac{\sqrt{pD}}{k_{1}S}tanh\left(\sqrt{\frac{p}{D}}h\right)}$$

The concentration profile *C* (*z*,*t*) of Zn(II) ions is, in principle, obtainable by applying the inverse Laplace transform to Equations (9) or (10). However, their forms are not trivial, and this operation cannot be readily resolved analytically. Therefore, we perform the inverse Laplace transform within a numerical approach using a MATLAB code based on [36,37].

To deduce the time-dependent axial growth rate of ZnO NWs, denoted as *R_c_*, we perform mass balance at the NW growth front, from which we can deduce the following relation:(11)$$R_{c}\left(t\right)=\frac{k_{1}}{\rho }\left(C\left(z=0,t\right)-C_{eq}\right)$$
where *ρ* is the atomic density of wurtzite ZnO that is equal to 4.20 × 10^28^ m^−3^. The length of ZnO NWs, denoted as *L,* and its temporal dependence is, in turn, deduced by integrating Equation (11):(12)$$L\left(t\right)=\int_{0}^{t}R_{c}\left(\tau \right)d\tau$$

Interestingly, *R_c_* and *L* can be deduced only from the concentration profile of Zn(II) ions at *z* = 0. To simplify the problem, we implement the numerical approach of the inverse Laplace transform of Equation (10) instead of considering Equation (9). To compute this integral from discrete values of *R_c_* taken at different times, *t*, a linear interpolation is performed between two consecutive values of *R_c_*. Furthermore, the value of the equilibrium concentration of Zn(II) ions, *C_eq_*, depends on the CBD conditions and is generally not negligible [33]. However, it can be typically determined from thermodynamic simulations, as detailed in [33]. Additionally, the homogeneous formation of ZnO taking place in the bulk of the chemical bath can strongly impact the growth kinetics of ZnO NWs heterogeneously formed from the substrate, particularly in the case of CBD performed in standard pH conditions [22,33,38]. As the massive precipitation attributed to homogeneous growth is commonly observed during the first stages of CBD [33], we consider, in a first approximation, that it occurs instantaneously at *t* = 0. As a consequence, the effective value of the initial concentration, *C_0_*, is usually considered to be lower than the concentration of precursors introduced in the chemical bath and needs to be deduced from experimental data. A more detailed description of this theoretical model is available in the Appendix A.

### 3.2. Comparison of the Theoretical Model with the Case of a Semi-Infinite Reactor

To check the consistency of the present theoretical model implementing a finite reactor height, *h*, with the model described in [33] considering a semi-infinite reactor, the evolutions of the ZnO NW length and axial growth rate with time were calculated for different reactor heights and compared to Equations (2) and (3) (taken from [33]). The theoretical curves obtained for *h* values ranging from 0.5 to 2 cm during a CBD performed at 90 °C with an initial concentration of Zn(II) ions of 30 mM are presented in Figure 2. Typical parameter values of 19.1 mM, 15.2 mM, and 18.2 µm/s were selected for *C*_0_, *C_eq_*, and *k*_1_, respectively, as determined in [33], whereas the Zn(II) ion diffusion coefficient *D* was deduced from [39] and set to 2.74 × 10^−9^ m^2^/s. Finally, a typical value of the *c*-plane top surface area ratio, *S*, of 0.3 was considered [33]. We can notice that when *h* values increase from 0.5 to 2 cm, the evolutions of both the ZnO NW length and axial growth rate become increasingly similar to the evolutions observed in the case of a semi-infinite reactor, whose curves act as asymptotes when the reactor height tends to infinity. These results show the great consistency between the two models.

Furthermore, Figure 2 reveals the strong influence of the reactor height on the growth kinetics of ZnO NWs. Indeed, the axial growth rate is shown to be significantly reduced for small *h* values and long growth times, which can typically be correlated with the depletion of reactants in the chemical bath. For instance, in the case of a small reactor height of 0.5 cm, the NW axial growth rate appears to become negligible—while the NW length reaches a constant value of 1 µm—after roughly 4 h of effective growth time, which can be related to a fully depleted chemical bath. It is, thus, possible to describe the evolution of the NW length with time according to three successive growth regimes: (i) when the effective growth time is smaller than a certain characteristic time, denoted as *t_h_*, the chemical bath is not significantly depleted in reactants and the elongation process of ZnO NWs follows similar kinetics as in the case of a semi-infinite reactor; (ii) when the effective growth time becomes larger than *t_h_*, the chemical bath depletes significantly faster than in the case of a semi-infinite reactor due to its limited height, which leads to a progressive decrease in the axial growth rate of ZnO NWs; and (iii) when the effective growth time is much larger than *t_h_*, the chemical bath is fully depleted and the NW growth is completely stopped.

To quantify with more precision the time, *t_h_*, from which the reactor height, *h*, becomes significantly influential on the axial growth rate of ZnO NWs, we propose to define it more rigorously as the effective growth time needed to reach a relative difference, *α*, between the lengths of ZnO NWs grown in a reactor of semi-infinite height, *L_∞_*, and of finite height *h*, *L_h_*. Therefore, *t_h_* is reached when ∆*L*/*L* = *(L_∞_—L_h_)/L_∞_* = *α*. With this criterion, *t_h_* was determined theoretically for different *h* values and for *α* values of 1%, 3%, and 5%, as revealed in Figure 3. The obtained values for *α* = 5% are also reported in Figure 2 (dashed lines). We observe that *t_h_* follows a parabolic evolution with *h*, as it rapidly increases for a long effective growth time—e.g., *t_h_* > 24 h, when *h* > 3 cm. Furthermore, the conditions in which the different NW growth regimes occur can readily be visualized from this graph: when *t* < *t_h_*, the reactor can be considered to be of semi-infinite height and the axial growth rate of ZnO NWs is maximized, whereas when *t* > *t_h_*, the reactor height must be taken into account in the determination of the ZnO NW length, as the NW growth becomes significantly slower. As a general rule, the value of *α* is typically chosen to reflect the maximum drop of axial growth rate that can be tolerated in a particular system. 

Correlatively, it is also possible to deduce from the theoretical model the growth time needed for the complete depletion of the chemical bath from which the ZnO NW growth becomes negligible. This value can be typically estimated by considering the time required for the ZnO NW axial growth rate to drop down to an arbitrary low value (e.g., in the range of 10–50 nm/h). In the growth conditions considered in Figure 2 and Figure 3, the theoretical model predicts that the NW growth will become negligible (i.e., *R_c_* < 50 nm/h) after 3.4, 9.3, 16.5, and 24.0 h for reactor heights of 0.5, 1.0, 1.5, and 2.0 cm, respectively. It is worth noticing that most of the reactors used experimentally have a height of a few centimeters—usually greater than 2 cm—and that effective growth times are only of a few hours—usually lower than 24 h. Therefore, *t_h_* is typically not reached, and the reactor can be approximated to a semi-infinite medium in the wide majority of cases in the present experimental conditions considered. However, the reactor height is expected to have a stronger influence in other non-standard, but important experimental conditions, such as when different chemical precursor concentrations [22,40,41], growth temperatures [41], or pHs [38] are used.

### 3.3. Comparison of the Theoretical Model with Experimental Data

To compare the theoretical model with experimental data, the growth of a series of samples was carried out where ZnO NWs were formed by CBD at 90 °C with 30 mM of Zn(NO_3_)_2_ and HMTA and with varying reactor heights, *h,* ranging from 0.5 to 4.0 cm. To be consistent with the geometry used in the theoretical model, the substrates were placed face down at the top of the chemical bath, as depicted in Appendix A. Furthermore, to obtain a noticeable variation of the length of ZnO NWs with the reactor height, a very long growth time of 41 h was chosen. The FESEM images collected on all the series of samples reveal a slight increase in the length of ZnO NWs as the reactor height is increased, as shown in Figure 4. The mean length of ZnO NWs for each reactor height was deduced from these FESEM images and is reported in Figure 5. The effect of the reactor height on the length of ZnO NWs was confirmed from this graph: the mean length of ZnO NWs increased from around 1.3 to 1.7 µm when the reactor height was increased from 0.5 to 4.0 cm.

This evolution was subsequently fitted with the theoretical model to check its consistency with the present experimental data. To do so, an effective growth time of 40 h and 25 min was considered, as a duration of around 35 min is typically necessary for the chemical bath to reach its final temperature of 90 °C. The values of the *c*-plane top surface area ratio, *S,* were also measured from top-view FESEM images on each sample of the series, as reported in Appendix A. They were found to be fairly constant around 0.30, which was, therefore, considered in the theoretical model. Accordingly, *C*_0_, *C_eq_*, and *k*_1_ were set to 19.1 mM, 15.2 mM, and 18.2 µm/s, respectively, as determined from [33], and *D* to 2.74 × 10^−9^ m^2^/s, as deduced from [39]. One can notice that *C*_0_ is considered to be lower than the concentration of chemical precursors introduced, which accounts for the initial consumption of reactants from homogeneous growth. Interestingly, when using the theoretical model with the effective growth time of 40 h and 25 min considered in the series of samples, the length of ZnO NWs was found to be greatly overestimated for reactor heights above 1 cm, as revealed in Figure 5 (black curve), whereas values reaching up to 4.1 µm were predicted for reactor heights above 3.5 cm. This apparent inconsistency between the theoretical model and the experimental data could be explained by the influence of the homogeneous growth of ZnO taking place in the bulk of the chemical bath, as this was not rigorously implemented in the present model. The homogeneous growth was indeed considered, in a first approximation, as an instantaneous phenomenon taking place at *t* = 0. While the massive precipitation attributed to homogeneous growth is mainly observed in the first stages of CBD in the present growth conditions [33], it is reasonable to assume that it can occur as well throughout the rest of the CBD until the chemical bath is fully depleted of reactants (i.e., when [Zn(II)] = *C_eq_*). This is supported by the fact that ZnO precipitates can typically be observed in the whole chemical bath even after several hours of growth [33]. As a consequence, homogeneous growth leads to a faster consumption of the chemical reactants where the complete depletion of the chemical bath occurs much sooner than the model predictions, thus significantly affecting the growth kinetics of ZnO NWs heterogeneously grown from the substrate. Additionally, it should be mentioned that spontaneous natural convection at the microscopic scale through eddy phenomena was not taken into account for the sake of simplicity, but may play a non-negligible role here as it is known to affect the diffusion of chemical species in the bulk of macroscopically still solutions [42,43].

Interestingly, when considering the effective growth time as the only free parameter in the theoretical model, a good fit can be obtained with the experimental data for a relatively short effective growth time of 7 h (blue curve). This result confirms that the faster depletion of reactants due to ZnO homogeneously grown in the bulk of the chemical bath is associated with a much lower effective growth time of ZnO NWs heterogeneously grown from the substrate. It can be noted that the mean ZnO NW length obtained for a reactor height of 0.5 cm remains slightly higher than the fit from the theoretical model irrespective of the effective growth time considered, which could be attributed to additional contributions such as the influence of natural convection. Therefore, although the theoretical model correctly predicts the general form of the evolution of the ZnO NW length with the reactor height, experimental data are still needed to precisely adjust the effective growth time and take into account the homogeneous growth of ZnO. 

### 3.4. Using the Theoretical Model as a Predictive Tool

The present theoretical model represents a valuable tool to predictively assess the influence of the reactor height and growth conditions on the growth kinetics of ZnO NWs. To illustrate this point, a 3D plot of the theoretical evolution of the length of ZnO NWs with the effective growth time, *t,* and the reactor height, *h,* for a chemical bath kept at 90 °C initially containing 30 mM of Zn(NO_3_)_2_ and HMTA is shown in Figure 6a. For the sake of clarity, the theoretical data are also presented on 2D graphs at given reactor heights, *h,* in the range of 0.1 to 5 cm (Figure 6b) and at given effective growth times, *t,* in the range of 0.5 to 24 h (Figure 6c). We can notice that the evolution of the length of ZnO NWs with the reactor height reflects the growth regimes previously described: (i) in the first stages of the growth—i.e., when *t* < *t_h_*—the length of ZnO NWs is completely independent of the reactor height and the ZnO NW axial growth rate is maximized; (ii) in the later stages of the growth—i.e., when *t* > *t_h_*—the length of ZnO NWs becomes noticeably dependent of the reactor height and the ZnO NW axial growth rate correlatively decreases; and (iii) in the lasts stages of the growth—i.e., when *t* >> *t_h_*—the length of ZnO NWs becomes highly dependent upon the reactor height and the ZnO NW axial growth rate becomes negligible.

The theoretical model developed here, implementing the reactor height in the NW growth kinetics, represents a big step forward toward the deeper understanding of the physicochemical processes at work during their elongation and the design of an effective batch reactor for industrial purposes. However, despite the progress mentioned, it also presents several limitations that need to be considered. Firstly, more precise implementation of the homogeneous growth of ZnO in the calculated NW axial growth rate is required and may be achieved for instance by directly modifying the diffusion equation, as described by Černohorský et al. [34], although this approach considerably increases the difficulty of the analytical solution and typically requires more complex numerical methods. Secondly, the occurrence of spontaneous natural convection at the microscopic scale is currently not implemented despite its contribution to the diffusion of chemical species [42,43]. Thirdly, to completely describe the influence of the reactor geometry on the ZnO NW growth, more realistic geometrical conditions may be considered through the implementation of finite lateral reactor and substrate sizes, which implies the solution of diffusion equations in two or three dimensions. Finally, the development of a universal model accurately describing the growth of semi-conducting materials by CBD compatible with a broad range of experimental conditions represents a stimulating challenge for upscaling purposes and fundamental studies.

## 4. Conclusions

In summary, we have established a theoretical model reporting the elongation process of ZnO NWs grown by CBD under dynamic conditions from a closed reactor of finite height, corresponding to the most typical configuration encountered in the investigations reported in the literature in which a finite volume of solution is progressively depleted in chemical reactants. The theoretical model is based on the solution of Fick’s diffusion equations in which the reactor height is implemented through boundary conditions. The time-dependent axial growth rate and elongation of ZnO NWs were deduced from this analytical approach, while the final step of the solution—involving the inverse Laplace transform of complex functions—was performed numerically. The influence of the reactor height on the ZnO NW elongation was assessed for a broad range of CBD conditions, which shows a relatively good consistency with experimental data although a shorter effective growth time should be taken into account due to homogeneous growth of ZnO occurring in the bulk of the chemical bath. Interestingly, the present theoretical model shows that the use of reactor heights smaller than 2 cm strongly reduces the ZnO NW axial growth rate in experimental conditions typically used—i.e., with effective growth times lower than 24 h—due to the faster depletion of reactants. Correlatively, by further comparing this approach with the model previously developed in dynamic conditions considering a semi-infinite reactor, it was found that reactors with heights of more than 2 cm could be approximated by a semi-infinite medium in the vast majority of the CBD conditions experimentally used. Finally, the present theoretical model represents a valuable tool that can be used as a predictive approach to design batch reactors for a wide variety of chemical precursors and semiconductor materials in applied research and industrial production.

## Figures and Tables

**Figure 1 nanomaterials-12-01069-f001:**
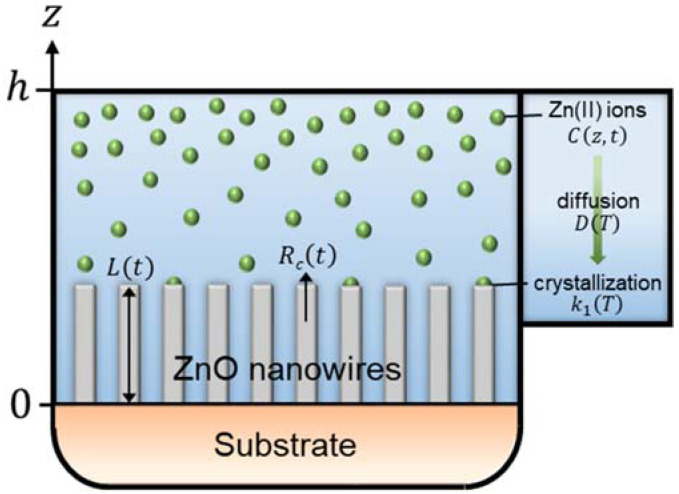
Schematic of the geometry used to establish the theoretical model describing the time-dependent axial growth rate and length of ZnO NWs by CBD under dynamic conditions, where a finite reactor height, *h*, is considered.

**Figure 2 nanomaterials-12-01069-f002:**
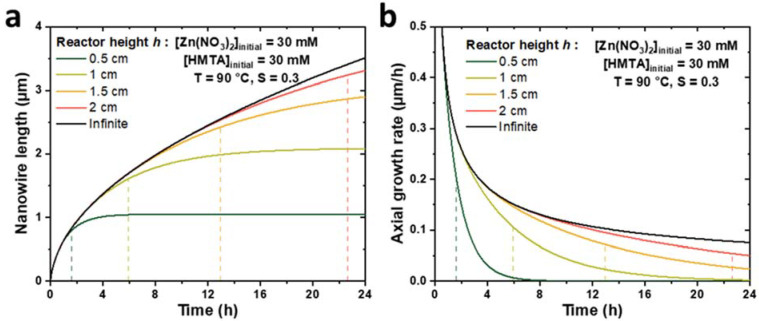
Theoretical evolutions of (**a**) the length and (**b**) the axial growth rate with the effective growth time of ZnO NWs for different reactor heights, *h*. For *h* values ranging from 0.5 to 2 cm (colored lines), the curves are determined by the numerical approach detailed in this work, whereas for an infinite *h* value (black lines), the curves are determined by Equations (2) and (3) (taken from [33]). The dashed lines represent the effective growth time *t_h_* (taken at *α* = Δ*L/L* = 5%, see text) for each reactor height, *h,* considered. The values taken for the different parameters correspond to typical growth at 90 °C with 30 mM of Zn (NO_3_)_2_ and HMTA (as determined in [33]), i.e., *C*_0_ = 19.1 mM, *C_eq_* = 15.2 mM, *k*_1_ = 18.2 µm/s, *S* = 0.3, *D* = 2.74 × 10^−9^ m^2^/s, and *ρ* = 4.20 × 10^28^ m^−3^.

**Figure 3 nanomaterials-12-01069-f003:**
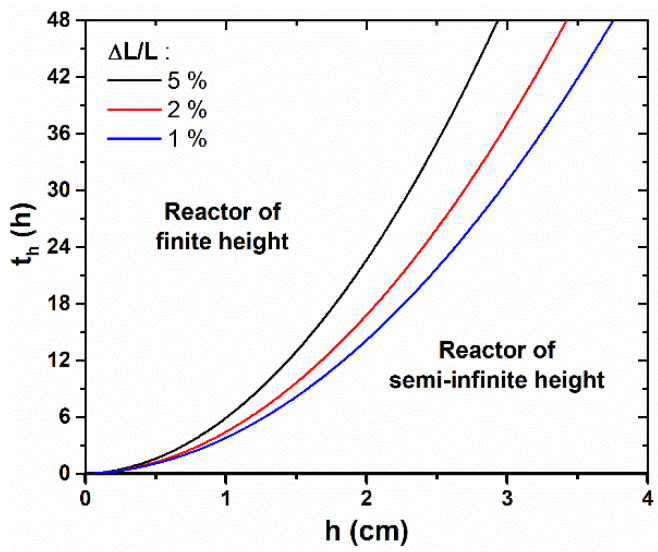
Theoretical evolution of the effective growth time, *t_h_*, from which the ZnO NW length, *L,* becomes significantly lower than the case where the reactor is of semi-infinite height, i.e., when Δ*L/L* becomes greater than 1%, 3%, or 5%, as a function of *h*. If the effective growth time does not exceed *t_h_*, the problem can be simplified to the case where the reactor is of semi-infinite height. The values of the different parameters correspond to a typical growth at 90 °C with 30 mM of Zn (NO_3_)_2_ and HMTA (as determined in [33]), i.e., *C*_0_ = 19.1 mM, *C_eq_* = 15.2 mM, *k*_1_ = 18.2 µm/s, *S* = 0.3, *D* = 2.74 × 10^−9^ m^2^/s, and *ρ* = 4.20 × 10^28^ m^−3^.

**Figure 4 nanomaterials-12-01069-f004:**
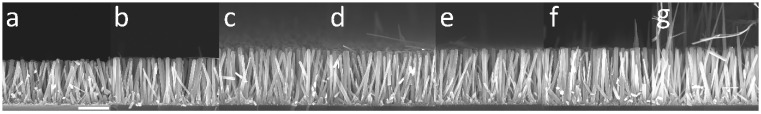
Cross-sectional-view FESEM images of ZnO NWs grown by CBD at 90 °C with 30 mM of Zn(NO_3_)_2_ and HMTA for 41 h with reactor heights of (**a**) 0.5, (**b**) 0.9, (**c**) 1.2, (**d**) 1.9, (**e**) 2.1, (**f**) 2.7, and (**g**) 4.0 cm, respectively. The scale bar represents 1 µm.

**Figure 5 nanomaterials-12-01069-f005:**
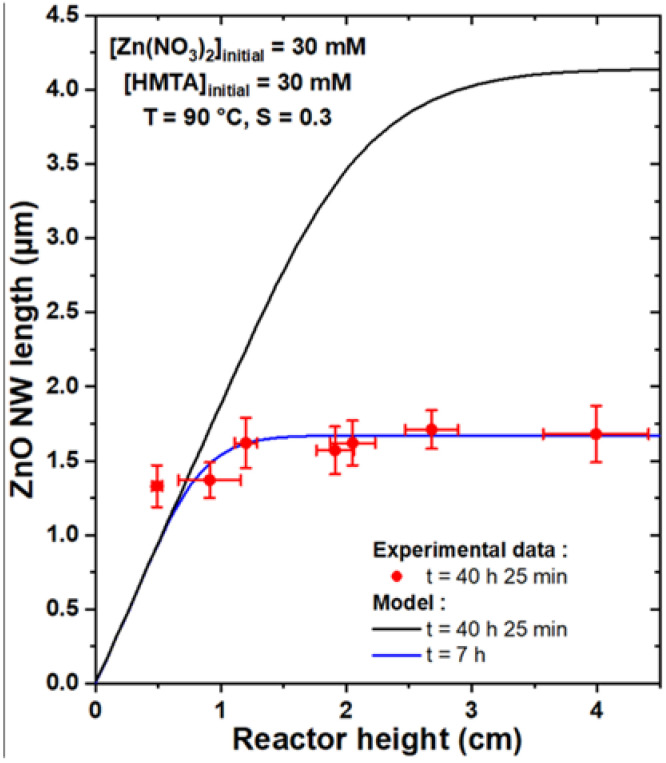
Length, *L*, vs. reactor height, *h*, of ZnO NWs grown by CBD at 90 °C with 30 mM of Zn (NO_3_)_2_ and HMTA for 41 h (corresponding to an effective growth time of 40 h and 25 min). The experimental data were fitted by the theoretical model, using *C*_0_ = 19.1 mM, *C_eq_* = 15.2 mM, *k*_1_ = 18.2 µm/s, *S* = 0.30, *D* = 2.74 ×10^−9^ m^2^/s, and *ρ* = 4.20 × 10^28^ m^−3^. The best adjustment is reached when an effective growth time of 7 h is considered, whereas the length of ZnO NWs is greatly overestimated if the expected effective growth time of 40 h and 25 min is instead considered.

**Figure 6 nanomaterials-12-01069-f006:**
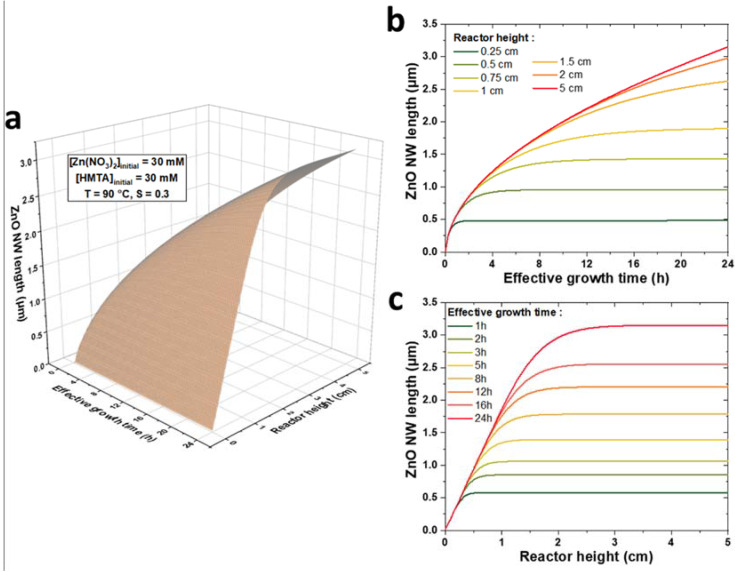
(**a**) 3D plot of the evolution of ZnO NW length vs. effective growth time, *t,* and reactor height, *h,* for a chemical bath kept at 90 °C initially containing 30 mM of Zn (NO_3_)_2_ and HMTA, as computed by the theoretical model using *C_0_* = 19.1 mM, *C_eq_* = 15.2 mM, *k*_1_ = 18.2 µm/s, *S* = 0.30, *D* = 2.74 × 10^−9^ m^2^/s, and *ρ* = 4.20 × 10^28^ m^−3^. Two-dimensional plots of ZnO NW length vs. (**b**) effective growth time, *t,* for specific *h* values and (**c**) reactor height, *h,* vs. specific *t* values are also provided.

## Data Availability

The data presented in this study are available on request from the corresponding author.

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
