# Peer review of "Implementing the Reactor Geometry in the Modeling of Chemical Bath Deposition of ZnO Nanowires"

_nanomaterials, 2022, doi:10.3390/nano12071069_

Round 1
Reviewer 1 Report
In this paper, the authors reported a theoretical model reporting the elongation process of ZnO NWs grown by CBD under dynamic conditions, where the reactor height is implemented through boundary conditions. The influence of the reactor height on the ZnO NW elongations was investigated, which showed a relatively good consistency with experimental data. After consideration, I can not recommend it be accepted for publication in Nanomaterials at present. The reasons were listed below:
- First, the authors claimed they reported a new theoretical description of the axial growth rate of nanowires in dynamic conditions as shown in Abstract. However, this theoretical description was already reported in the authors’ published work (J. Phys. Chem. C 2019, 123, 48, 29476–29483). The Figure 1 in this work was almost the similar with that in their published work. And most equations were almost similar.
- The main difference between this work and published work was that reactor height was assumed as finite value in this work but semi-infinite value in published work. Therefore, this work was just a study on the influence of one variable parameter of reported model. The novelty was not sufficient for Nanomaterials.
- From the results, it could be seen from Figure 5 that the influence of reactor height was more obvious when it was below 2um. However, as mentioned by the authors, most of the reactors used experimentally have a height of few centimeters – usually greater than 2 cm. Therefore, I think that the significance of this model used as prediction tool for the experiment is very limited.
Reviewer 2 Report
In this manuscript, the authors present a very careful and thorough calculation about the influence of reactor dimensions on diffusion-limited growth starting at one face of the reactor. The procedure is well described and the conclusions are valid under the chosen boundary conditions, but one important, often even decisive, phenomenon has been neglected: natural convection. To discuss, however, diffusion-limitation over centimeter distances appears meaningless since natural convection driven by density-differences as a consequence of chemical reactions is limiting any realistic diffusion-layer to thicknesses smaller than about 1 mm. This is well known from classic (electro-)chemical kinetics (e.g. book of K.J.Vetter) as well as from more recent studies (e.g., https://doi.org/10.1016/S0022-0728(00)00378-8, https://doi.org/10.1016/j.cis.2008.01.007). Such upper limitation of the diffusion layer thickness by external factors hinders a useful description of the influence of reactor size by mere consideration of diffusion as single process of mass transport as intended by the authors. Aside from this principle problem, a few minor problems were detected. Major revision of the work is needed before publication can be recommended. The following comments may help in this process.
- As indicated above as the main problem of this manuscript, natural convection cannot be neglected, in particular at a temperature of 90 °C. The strong deviation of theory and observation (Fig. 5) may well be caused by this neglection.
- The assumption of homogeneous crystallization as main reason for such deviation appears weak since after some time the precipitate should sink towards the bottom of the reactor and its further growth should lead to less pronounced changes for higher reactors (increased distance to sample).
- It is not clear to this reviewer why the authors did not add ammonia (as in their earlier work) to suppress the homogeneous precipitation reaction.
- A closer look at the experimental data in Figure 5, further, indicates deviation of the results even for the smallest reactor from the expected characteristics. In fact, the error bars indicate that even the model of infinite diffusion would not explain the high length found for this reactor. An independent parameter (natural convection?) may influence the model and, hence, weaken its predictive power. Further, the origin of error margins should be indicated as well as an information on the number of runs that the statistics are based upon.
- The final length of nanowires seems to be an inappropriate parameter to test the model. Since time-dependence is key in the model, time-dependent length measurements would be appropriate as performed, e.g., in the author´s earlier work. Reactions over hours are easily monitored and even the delay time of 35 minutes should then come out as a result rather than an assumption.
- To model the growth of ZnO nanowires by 1D-diffusion seems inappropriate for transport distances of centimeters given the considerable influence of 3D-diffusion already noticed on the µm scale (e.g., doi:10.1021/acs.jpcc.9b08958).
- A predictive power of the model cannot be claimed since the present results are not at all reproduced by the model. Only by help of an additional parameter (non-suppressed homogeneous crystallization) could an explanation be found. By no prediction, however, the time of 7h instead of the real 41 h could have been chosen. Therefore, the predictive power is very low. The data used in this section could easily be used in Figure 2a (Figure 6b) or Figure 5 (Figure 6c).
- The almost constant length of nanowires found after reaction in the differently high reactors (Figure 4) does not allow the conclusion that the reactor height is of decisive influence (Line 370).
Minor aspects:
- Figure 1 is an almost complete copy from an earlier paper of the authors. Either it should be cited from there or a new Figure should be used that emphasizes the present focus. In more than one part of the work, phrases are repeated almost literally from earlier work [33] or even from other parts of the present work. Such practice should be avoided. E.g., large parts of the Supporting Information are literally repeating parts of the manuscript. Identical equations are used and are even given new numbers. Such practice is confusing the readership rather than supporting it. It is appreciated if intermediate steps and additional arguments and hints are included here, but not at the cost of literal repetitions.
- According to the experience of this reviewer, “solution” of a differential equation is a more appropriate term than “resolution”.
- Typo “reasearch” in the last line of the abstract.
- No units need to be assigned to the parameters in the equations since no numbers are included. The equations work for parameters in any unit.
- The source and treatment (thermal oxide?) of Si wafers, the composition of the controlled atmosphere (air, inert gas, oxygen?) as well as source and purity of solvents (incl. water) should be included in the experimental part since crystal growth can be delicately influenced by impurities.
- In Line 192, it is unclear to which paper (present or [33]) the equation numbers refer to. In the caption of Figure 2, they seem to refer to [33].
Round 2
Reviewer 1 Report
The authors have addressed my questions. I think it can be accepted for publication.
Reviewer 2 Report
The authors have dealt with my comments in a very appropriate way and have revised their manuscript accordingly. I recommend publication of the work as is.